# Possible Interdisciplinar Standard for the Care of Pregnant Women Living with HIV-Polish Experience

**DOI:** 10.3390/healthcare10101949

**Published:** 2022-10-05

**Authors:** Katarzyna Plagens-Rotman, Piotr Merks, Magdalena Pisarska-Krawczyk, Witold Kędzia, Jaskulska Justyna, Magdalena Czarnecka-Operacz, Grażyna Jarząbek-Bielecka

**Affiliations:** 1Center for Pediatric, Adolescent Gynecology and Sexology Division of Gynecology, Department of Perinatology and Gynecology, Poznan University of Medical Sciences, 61-758 Poznan, Poland; 2Department of Pharmacology and Clinical Pharmacology, Faculty of Medicine, Collegium Medicum, Cardinal Stefan Wyszynski University in Warsaw, 01-938 Warszawa, Poland; 3Nursing Department, President Stanisław Wojciechowski State University of Kalisz, 62-800 Kalisz, Poland; 4Division of Gynecology, Department of Perinatology and Gynecology, Poznan University of Medical Sciences, 61-758 Poznan, Poland; 5Higher School of Strategic Planning in Dąbrowa Górnicza, Kościelna 4, 41-303 Dąbrowa Górnicza, Poland; 6Allergic and Occupational Skin Diseases Unit, Department of Dermatology, Medical University of Poznań, 60-355 Poznan, Poland

**Keywords:** HIV, AIDS, COVID-19 pandemic

## Abstract

HIV data for 2020 show a decline in the number and rate of new HIV cases diagnosed in the EU during the last decade. The COVID-19 pandemic has paralyzed the functioning of healthcare facilities in Poland and worldwide, also impacting the detection of HIV infection. Early diagnosis of HIV and implementation of antiretroviral treatment limit HIV transmission. A woman with HIV diagnosed during pregnancy should be under the care of a specialist in infectious diseases experienced in antiretroviral treatment. In this way, she will be properly protected during the delivery, and relevant medications can be implemented for the newborn baby. Taking these aspects into account, the medical team should consist of: A specialist in infectious diseases, an obstetrician, a neonatologist and pediatrician, a midwife, and a dermato-venereologist. Every effort should be made to increase the scope and quality of monitoring of the spread of the epidemic in Poland, with special emphasis on diagnostics based on specific tests among populations particularly exposed to HIV infections cooperating with non-governmental organizations.

## 1. Introduction

Human immunodeficiency virus (HIV) is a retrovirus belonging to the genus Lentivirus, is spherical and surrounded by a lipoprotein membrane with anchored transmembrane glycoprotein gp41 and gp120 exterior envelope glycoprotein. The HIV genome consists of two identical single-stranded RNA molecules and contains three genes typical of retroviruses (gag, pol, env), and regulatory genes (tat, rev, nef, vpu, vpr) [1,2,3,4].

HIV enters the cell through the CD4 receptor and co-receptors (CCR5 and CXCR4 are the best-known so far). The binding of gp120 with CD4 and a co-receptor triggers the fusion of the viral envelope with the cell membrane and leads to the nucleocapsid entering the cytoplasm and being uncoated. The release of the viral genes and enzymes begins reverse transcription (of viral RNA into pro-viral DNA), resulting in DNA being transported to the cell nucleus and integrated with the host genome.

In the clinical picture, primary infection can resemble infectious mononucleosis, often an oligosymptomatic stage with a low-grade fever for a few days. The acute retroviral syndrome is suspected if patients report risky sexual behaviors or exposure to blood-borne pathogens over the last few weeks, “mononucleosis” has reoccurred in an adult patient, or the symptoms of other sexually transmitted diseases (e.g., syphilis, gonorrhea) have been observed. An asymptomatic phase is found during the primary viral phase as a consequence of a relative balance between HIV replication and the immune response to infection. In people who do not take antiretrovirals, this phase can last between 1.5 to 15 years [5]. Persistent generalized lymphadenopathy occurs in a large proportion of patients in the late asymptomatic phase and is characterized by enlarged lymph nodes (>1 cm in diameter) in >2 or more areas apart from the groin, chronic fatigue for 3 or more months, headaches, enlarged spleen (~30%), and more frequent infections of the skin, respiratory tract and alimentary tract caused by non-opportunistic microorganisms. The number of CD4 cells gradually decreases (on average by 30–90/mm^3^ per year) [6,7], and the pace of this DOWNWARD TREND is dependent on the initial viral load, i.e., the higher the viral load, the faster the reduction in CD4 cells.

In the symptomatic phase, a development is observed in symptoms of opportunist infections due to a reduced number of CD4 lymphocytes. The clinical course is usually mild, and the most common diseases are herpes zoster, bacillary angiomatosis (that appears as erythematous skin lesions resembling those of Kaposi sarcoma and caused by Bartonella henselae), hairy leucoplakia (in the form of eruptions similar to oral candidiasis, mainly on the lateral border of the tongue), candidiasis of the throat or vulvovaginal candidiasis (persistent, recurring or refractory), cervical intraepithelial neoplasia and carcinoma in situ of the cervix (HPV infection), fever for one or more months, chronic diarrhea for one or more months, clinical symptoms of thrombocytopenia, listeriosis, peripheral neuropathy, and pelvic inflammatory disease.

AIDS is the final stage of HIV infection. Apart from indicator diseases in the form of opportunist infections or cancers, the diagnostic criterium for AIDS is the gradual decrease in CD4 lymphocyte cells in peripheral blood (below 200 cells/mm^3^).

It was estimated that at the end of 2020, 37.7 million people were living with HIV, 25.4 million of whom were in Africa, 680,000 of whom died from HIV-related causes, and 1.5 million of whom had acquired HIV [8]. Every effort should therefore be made to achieve the global 95-95-95 targets launched by the Joint United Nations Program on HIV and AIDS to avoid the worst-case scenario of half a million HIV-related deaths in sub-Saharan Africa, increases in HIV infections due to limited access to HIV-related medical services during COVID-19, and the slowing of public health responses to HIV [8].

The COVID-19 pandemic has paralyzed the functioning of healthcare facilities in Poland and worldwide, also impacting the detection of HIV infections. In 2020, there were 934 newly confirmed cases of HIV infections in Poland, i.e., 53% less than in 2019 (1763 cases) [9]. Similar situations were observed in Italy, Spain, and Taiwan [1]. Lower detection rates of HIV infections in Poland at the beginning of the pandemic were probably associated with the limited activities of consulting and diagnostic facilities.

It should be borne in mind that the length of the pandemic and the resulting restrictions may have also contributed to more frequent risky sexual behaviors and psychoactive substance consumption [9].

HIV (human immunodeficiency virus) and AIDS (acquired immune deficiency syndrome) are major worldwide public health issues in Poland. In 2019, there were 25,591 new HIV infection cases registered in Poland [10], but it should be emphasized that there may be only about 20,000 infected people [9], so discrepancies should be investigated to standardize and improve the functioning of the epidemiological surveillance system, and to provide coordinated cooperation between clinicians and institutions performing epidemiological surveillance. For this reason, the recommendations of the Polish Scientific AIDS Society should be emphasized. They pay attention to:effective identification of reports received from healthcare entities via the patient identifier (patient’s initials + date of birth + gender + the last number of pesel (personal id. no.)),electronic reporting of new cases of HIV by laboratory managers, as well as physicians diagnosing HIV and certifying deaths of HIV+ persons,combining information from available databases, developing the scope of information in order to monitor the continuum of care of HIV patients in Poland, and the impact of migration to HIV epidemiology,full transparency in determining national recommendations and using them for national and international monitoring [9].

The COVID-19 pandemic largely hampers the continuation of extensive HIV care at all stages [11]. It should also be noted that there is a risk that, unfortunately, healthcare professionals may experience negative physical and psychological effects of the pandemic in the form of exhaustion.

The objective of this paper was an attempt to give an overview of the importance of the obstetric care of pregnant women with HIV.

## 2. Methods

Two independent reviewers searched medical and public databases, e.g., PubMed, Google Scholar, MEDLINE, using such search and MeSH terms as “HIV”, “COVID-19”, “rekomendacje” (recommendations), “ciąża” (pregnancy), “noworodek” (newborn). The inclusion criterium was to be an article published in a reviewed journal in the last 10 years (2012–2022). There were no limitations with regard to the publication language or the kind of studies.

In the article, the authors consider the aspects of caring for a pregnant woman, giving birth to a newborn mother with HIV (+), and the related recommendations.

## 3. The SARS-CoV-2 Pandemic vs. HIV as a Specific Interdisciplinary Issue

The COVID-19 pandemic has paralyzed the functioning of healthcare facilities in Poland and worldwide, also impacting the detection of HIV infections.

Considering an interdisciplinary approach to HIV patients, special emphasis should be placed on the fact that an inappropriate way of informing patients about infections (including prognosis, possible treatment, and quality of life with HIV) has been practiced to date [6]. Van Haastrecht et al. [7] showed that a proper transfer of information about HIV/AIDS prevented suicidal attempts and overdosing on psychoactive substances among addicted people, especially intravenously. A proper transfer of information about an infection that reduces patient stress is extremely difficult and requires the education and experience of physicians in this area. Patients should be made aware of:the differences between HIV and AIDS,the efficiency and necessity of taking antiretrovirals and the resulting side effects (early and late),the ways of monitoring treatment effectiveness, andthe benefits of appropriate and systematic adherence to medications.

The initial information about a positive HIV diagnosis causes enormous stress, shock, strong emotional strain, a sense of helplessness resulting in frustration, disappointment, mood and behavior disorders, a loss of motivation, and strong anxiety symptoms. It is estimated that about 20–30% of patients with HIV are diagnosed with depression, and 50% with depressive symptoms [5], which mostly concerns women with HIV. It should be strongly emphasized that undiagnosed and untreated symptoms of depression remain in a significant correlation with the disease progress and three times greater risk of non-adherence [12]. As mentioned above, patients diagnosed with HIV are more likely to experience increased anxiety (over 45%) compared to the healthy population [13], with symptoms of a general nature. For this reason, before the formal diagnosis, it is important to take a detailed history of the patient, and their family/close people, and to conduct a psychiatric consultation as a significant element of the comprehensive care of patients with HIV. Thus, the cooperation of an interdisciplinary team of specialists is necessary in these cases [12,14].

The issues related to the sexual life of patients infected with HIV are rarely the subject of discussion among specialists in this area of medical science. This also concerns dermato-venereologists and specialists in infectious diseases in Poland, which is probably connected with the awkwardness and insufficient knowledge of human sexuality [15]. Similar determinants of insufficient communication in terms of the above-mentioned problem are observed in the patient population.

Discussing issues related to the sexuality of people diagnosed with HIV is necessary to reduce the mental side effects of the situation. A global positive effect of such activities can be achieved by improving patient quality of life and influencing conscious decisions regarding sexual activity and procreation. It is therefore recommended [16,17,18]:•to perform a detailed physical examination encompassing the sexual life, including:✓sexual orientation;✓the meaning and place of sexual contact in the patient’s life;✓the characteristics of sexual behaviors;✓the assessment of possible dysfunctions in sexual behaviors;•to pass on knowledge of established and recommended principles of infection prevention;•to educate patients in:✓the routes of infection transmission,✓sexual activity and the related risk of transmission of HIV to a partner,✓the ways of minimizing the risk of HIV infection through sexual contact,✓the necessity of informing a partner of one’s serological status, and✓the principles of reliable contraception.

The interdisciplinary team of specialists taking care of HIV patients also includes a dietician whose task is to modify patient lifestyle by changing their eating habits and physical activity [19]:eating 4–6 meals daily,energy intake of 30–35 kcal/kg body weight,protein intake of 1.5 g/kg body weight (1/2 animal protein, e.g., lean meat, milk and dairy products, fish),30% of daily energy requirements in the form of fats from butter, cream, milk, and oils,eliminating animal fat products, i.e., lard, pork fat, and fat,55–60% of daily energy requirements in the form of carbohydrates,intake of micronutrients covering 100–150% of recommended daily allowance.

It should be clearly stated that antiretroviral treatment in those with HIV allows them to build relationships with HIV-positive people and plan to have children. It also contributes to reducing the risk of transmission of the infection to sexual partners, maintaining an unchanged serological status of the couple, and thus systematically increasing the number of serologically different couples (HIV+/HIV−). Considering the stigmatization of HIV infections, partners not infected with HIV should be taken into specialist medical care in HIV outpatient clinics to receive advice on the prevention of HIV infection, HIV diagnostics, and family planning.

## 4. HIV in Pregnant Women = As the Special Obstetric Situation

Pregnant women are potentially more susceptible to severe viral infections due to a shift from cellular to humoral immunity during pregnancy. Some authors suggest a lower HIV incidence and mortality following COVID-19 infection in pregnant women in relation to the general population. Human chorionic gonadotropin and progesterone probably decrease Th-1 proinflammatory activity by reducing tumor necrosis factor alpha. For this reason, complex immunological modulation can have a protective influence prior to the cytokine storm on the resulting incidence and mortality due to COVID-19 among pregnant women [20]. In addition, pregnant women and newborns did not show a higher risk of death compared to the general population, and the causes of death among women was mainly related to previous comorbidities, and among newborns to pre-term births.

Lumbreras–Marquez et al. indicate that the most common symptoms of COVID-19 in pregnant women were fever, cough, dyspnea, fatigue and muscle pain. 95% of them did not manifest any symptoms or experienced mild symptoms, and 0.8% required intensive care. The most frequent comorbidities were gestational diabetes and obesity, which was also confirmed in studies by Lumbreras–Marquez et al. [21] and Takemoto et al. [22], where diabetes and obesity in the mothers caused a three-fold increase in mortality risk. Data for 2021 should be treated as temporary data to be updated, as due to the COVID-19 pandemic, data regarding the supervision over infectious diseases are reported with a delay.

It should be strongly emphasized that data on COVID-19 mortality and incidence among mothers are still insufficient and therefore greater knowledge is required in order to identify pregnant women with a higher risk of severe COVID-19 [23].

Data indicating the risk of COVID-19 and its influence on pregnant women and children with HIV are also insufficient. The higher risk can result from decreased immunity in these patients, and HIV+ pregnant women are more susceptible to rapid health deterioration following COVID-19 because of physiological and psychological susceptibility [24,25].

It should be highlighted that reducing the risk of transmission of infection during the period of pregnancy and delivery and also striving for improvement of life quality of women contributed to conscious planning offspring. In the light of contemporary knowledge and according to recommendations in the scope of organization and care of pregnant women, bearing a woman a newborn baby infection with HIV is not a contraindication against having offspring. Furthermore, pregnancy does not have a negative influence on the course of infection with HIV.

Pregnancy of a woman with HIV+ is an indication of the application of antiretroviral therapy, independently of the number of lymphocytes CD4 and concentration of HIV RNA. During the period of pregnancy, the kind of applied medicaments has to take into consideration the safety of the fetus and changes in pharmacokinetic parameters.

In 2016, the availability of prophylaxis for vertical transmission of HIV for pregnant women amounted to 76%.

According to the Ordinance the of Minister of Health of 16 August 2018 on the organizational standard of perinatal care (Journal of Laws of 2018, position 1756) each pregnant woman should the recommended examination for HIV twice. The first time should be in the first trimester (up to 10 weeks) and the second time should be in the third trimester of pregnancy (between 33 and 37 weeks of pregnancy). Simultaneously it is necessary to emphasize that the physician/midwife, i.e., person who provides health service is obliged to recommend examination for infection with HIV for a patient.

It is necessary to remember that proper prophylaxis contributes to decreasing the average risk of infection with HIV mother–child from 25–30% to <1% [26]. Pregnant women, with whom infection with HIV during pregnancy was diagnosed, should be under the care of a specialist for infectious diseases with experience in the scope of antiretroviral therapy, ensuring implementation of ART at woman, securing woman during delivery, and ensuring proper prophylaxis of new-born baby in the range of medicaments. Thus, taking these aspects into consideration, the medical team should be:specialist for infectious diseases,obstetrician,neonatologist and pediatrist,midwife, anda dermato-venereologist.

A characteristic feature of this medical team should be precisely coordinated and planned care in units with degrees of reference that take care of a pregnant woman who is infected with HIV, contributing to minimizing the risk of vertical transmission of HIV.

Best efforts should be used to increase the scope and the quality of monitoring the development of epidemics in the Polish population within the framework of cooperation of non-governmental organizations, especially taking into account testing populations that are exposed due to infection with HIV.

In a stressful situation such as a global pandemic, health care providers need to play a pivotal role to ensure pregnant individuals feel supported and receive consistent care throughout the pregnancy and postpartum period [27].

## 5. Recommendations for a Period of Pregnancy [28,29,30]

It is necessary to offer examination for HIV for all pregnant women during the first visit to a physician-obstetrician (noting conducting an examination or refusing examination in maternity notes and in medical documentation),A positive result (HIV+) should be communicated to the patient and attending physician no later than two weeks after conducting an examination,It is necessary to quickly ensure for pregnant women HIV+ consultation with a physician-specialist of infectious diseases in order to begin therapy,A pregnant woman who is addicted to psychoactive substances should be additionally subject to the care of a physician-psychiatrist (ensuring substitution therapy) and social worker,A center that takes antepartum care sends pregnant woman HIV+ to section with III reference degree in the 36th week of pregnancy,At pregnant women HIV+ examinations for other sexually transmitted infections should be conducted at the beginning of pregnancy and repeated in 28 weeks of pregnancy, including viral hepatitis C and B,It is necessary to offer for women who report to delivery without the result of examination for HIV to conduct a rapid test before admission labor ward, especially in the case of women who did not give consent to such examination earlier.

## 6. Monitoring and Prophylactic Antiretroviral Therapy (ARV)

Decision of beginning antiretroviral therapy and the way of applying and monitoring it during pregnancy is taken by a physician for infectious diseases who specializes in the therapy of women who live with HIV, according to recommendations of the Polish Scientific Society of AIDSDuring pregnancy monitoring therapy ARV (control of a number of CD4 and viremia (VL every 12 weeks). The last examination was in 36. week of pregnancy.The World Health Organization recommends that all pregnant and breastfeeding women with HIV irrespective of CD4 cell count, viral load, and clinical stage should have triple antiretroviral drugs, which should be maintained throughout the period of the risk of MTCT (late pregnancy, labor, and breastfeeding) and continued for life as for other patients with living HIV [31].

## 7. Care during Delivery and in the Postpartum Period

•Taking care of bearing woman HIV+ by medical personnel after earlier becoming acquainted with results of examinations,•Indications to Caesarean section, fixed by the Polish Gynaecological Society are the same as in the case of women who are not infected with HIV,•Caesarean section for women with HIV+ should be carried out for all women who obtain combined antiviral therapy with a detectable level of viremia (>50 copies RNA of HIV/mL),•Elective caesarean section should be performed in patients with a viral load ≥400 copies of HIV RNA at 36 weeks of gestation [28,32].•Planned caesarean section should be considered in pregnant women with a viral load of 50–399 copies of HIV RNA at 36 weeks of pregnancy.•Vaginal delivery may be considered in pregnant women with a viral load <50 copies of HIV RNA at 36 weeks of pregnancy and in the absence of obstetric complication•Elective Caesarean section should be considered at:✓Coexistence of infection with HIV and HCV, and✓Treatment with highly active antiretroviral therapy (HAART),•Avoiding invasive obstetric procedures (amniotomy, sampling pH from fetal scalp, episiotomy, obstetric forceps, and vacuum extractor) during delivery with forces of nature,•Medicating a child with antiretroviral medicaments—the exact time of beginning prophylactic therapy ARV should be noted in medical documentation,•Treatment of new-born babies should be adapted to recommendations of the Polish Scientific Society of AIDS and earlier arrangements of physicians who took prepartum care,•It is necessary to conduct examinations on new-born babies according to recommendations of the Polish Scientific Society of AIDS within the first 5 days of life,•Avoid breastfeeding.

## 8. Conduct with a New-Born Baby of a Mother Infected with HIV

•Exact washing new-born baby as soon as possible after birth,•Procedure of siphoning off an upper respiratory tract of the new-born baby of amniotic fluid and other secretions of the mother,•Implementation of prophylaxis in the first 24 h from the moment of delivery,•Avoiding breast-feeding [33,34],•Vaccination against viral hepatitis, in the case of children of mothers with HBsAg (+) together with immunoglobulin anti-HBs (within 12 h),•Absolute contraindication to vaccination with BCG,•Laboratory tests (the first 5 days of life of a new-born baby):✓peripheral blood count with differential count,✓marking the number of lymphocytes CD4 and CD8,✓viral culture, PCR HIV.•Further treatment of newborn babies in a reference center.

## 9. Conclusions

It should be borne in mind that the period of isolation due to the COVID-19 pandemic does not translate into a lack of support for those with HIV. The “Be With Us” information materials carefully prepared and promoted by the Association of Volunteers Against AIDS are of great value, as properly designed and conducted education is the best way of “fighting” any viral infections. Free effective and safe medications in the HIV-related area are available in Poland during the COVID-19 pandemic. On one hand, specialists in infectious diseases are responsible for treating the above-mentioned HIV+ population, while on the other hand, patients should remember this and adhere to any instructions.

Global organizational structures such as regulatory bodies for scientific research, ethical committees, and review committees, are continuously involved in preparing and updating guidelines regarding scientific, research, diagnostic, and therapeutic principles related to COVID-19 and HIV infections, as this is an extremely important global medical and social issue of the contemporary world.

Standards of the care of HIV+ pregnant women and newborns will enable performing all care activities aimed at improving women’s health. During pregnancy, women should enjoy compassionate continuous care by their families and friends, and primarily specialist care by medical staff. Many diseases that coexist during pregnancy create high-risk pregnancies, e.g., human immunodeficiency virus, which is now considered a chronic disease, the course of which can be significantly slowed down by appropriate treatment. An effective therapy requires cooperation between medical staff and patients. Care for HIV patients, contrary to many other infections, is considered and taken not just in the medical, but also to a large extent the psychosocial context.

## Data Availability

Not applicable.

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
