# Peer review of "Possible Interdisciplinar Standard for the Care of Pregnant Women Living with HIV-Polish Experience"

_healthcare, 2022, doi:10.3390/healthcare10101949_

Round 1

Reviewer 1 Report

Dear Authors,

The presented study tackles an important issue of HIV as an Interdisciplinary Clinical Problem for the Obstetric. I think that the article should be published but after major revision.  I get the impression that it’s rather a letter to editor than review article. The Introduction is rather the review of HIV pathology (there is nothing about pregnancy), and Results section looks like a plan of perinatal care in women affected with HIV in Poland. I think that more suitable title would be „ Possible Interdisciplinar Standard for the Care of Pregnant Women Living with HIV- Polish Experience”.

Some issues require complementary information:

1.       There is some typind error in author’s affiliations (It should be in sequence)

2.       The abstract should follow the style of structured abstracts, with or without headings: 1) Background: Placed the question addressed in a broad context and highlight the purpose of the study; 2) Methods: Described briefly the main methods or treatments applied; 3) Results: Summary of the article's main findings; and 4) Conclusion: Indicaton of the main conclusions or interpretations. The abstract should be an objective representation of the article.

3.       Methods- I suggest including information about selection of the articles method, data included, ecluded etc. ( See PRISMA)

4.       Verse 124- there is no information how did COVID-19  indluenced on detection of HIV in pregnant women (This is the title of the article). You should include statistics (how many pregnant women with HIV you have per year). Moreover there is no information how does it looks like now in Poland during COVID Pandemy- if there i sany difference.

5.       Overall – You should decide if you prefer to show your experience or make a review article with all requirements for that.

Author Response

Thank you very much for the reviews.

All corrections in the text are marked in yellow.

Authors

Reviewer 2 Report

Dear Authors 

I read the paper: "HIV as an Interdisciplinary Clinical Problem for the Obstetric", which falls whithin the aim of Healthcare. Honestly, the topic is interesting enough to attract the readers' attention, but the review would benefit from a comprehensive, minor revision. I have some specific recommendations around the review' and reporting that I describe in each section:

1) TITLE: "HIV as an Interdisciplinary Clinical Problem for the Obstetric".

However, the authors also explore the covid-19 era and HIV infection (as reported in paragraph 3). The title is not complete for this paper.

INTRODUCTION

2) I advise moving the HIV description (line 66) to the beginning of the introduction section: "Human immunodeficiency virus (HIV) is a retrovirus belonging..."

3) I propose a comprehensive explanation of the surveillance system in use in Poland., line 57.

4) The paper lacks data on the impact of COVID-19 on pregnancy. Consequently, I propose a concise summary of the covid's effect on pregnancy, citing the following paper:  

- https://doi.org/10.1002/ijgo.13726

- https://doi.org/10.1080/14767058.2020.1806817

5) At the end of the introduction, I recommend elucidating the goal of this review ( not reported).

6) Line 125: "Lower detection rates of HIV infections in Poland at the beginning of the pandemic were probably associated with the limited activities of consulting and diagnostic facilities, and later with the so-called epidemiological regime."

Covid -19 impacts all aspects of obstetric care, so I recommend enhancing this section by citing recent research: 

-https://doi.org/10.1111/jog.15205

https://doi.org/10.1016/j.ajog.2021.05.014

7)line 181: "the possibility of superinfection or REINFECTION ?????"

why this "????"

I suggested performing the minor revisions.

Author Response

(The authors gave the same response as above.)

Round 2

Reviewer 1 Report

Dear Authors

You have done a great job. The article looks much better. However I have some more suggestions:

1.       Abstract- I suggest deleting verses 27-39 and including the most important part or new ideas from Polish standard in Pregnant Women Living with HIV.

2.       Verses 45-63 I suggest shortening that part by 50%. We don’t need such specific characteristic of HIV in that subject.

3.       Verses 93-94 I suggest including this verses as the first sentence in verse 102

4.       Verses 119-122 I suggest changing the order of the sentences and starting with “The COVID-19 pandemic largely hampers the continuation of extensive HIV…….”

5.       Verse 121 I suggest including also one sentence how pregnant patient may feel in COVID pandemic

6.       Verses 137-148 I suggest including that part in Introduction.

7.       Verses 220-244 I suggest including that part in “4. HIV in pregnant women = as the special obstetric situation”

Author Response

(The authors gave the same response as above.)
